# Programming Puzzles

**Tal Schuster**
MIT

**Ashwin Kalyan**
Allen Inst. for AI

**Oleksandr Polozov**
Microsoft Research

**Adam Tauman Kalai**
Microsoft Research

## Abstract

We introduce a new type of programming challenge called programming *puzzles*, as an objective and comprehensive evaluation of program synthesis, and release an open-source dataset of Python Programming Puzzles (P3).[1] Each puzzle is defined by a short Python program $f$, and the goal is to find an input which makes $f$ return `True`. The puzzles are objective in that each one is specified entirely by the source code of its verifier $f$, so evaluating $f$ is all that is needed to test a candidate solution. They do not require an answer key or input/output examples, nor do they depend on natural language understanding. The dataset is comprehensive in that it spans problems of a range of difficulties and domains, ranging from trivial string manipulation problems, to classic programming puzzles (e.g., Tower of Hanoi), to interview/competitive-programming problems (e.g., dynamic programming), to longstanding open problems in algorithms and mathematics (e.g., factoring). We develop baseline enumerative program synthesis, GPT-3 and Codex solvers that are capable of solving puzzles—even without access to any reference solutions—by learning from their own past solutions. Codex performs best, solving up to 18% of 397 test problems with a single try and 80% of the problems with 1,000 tries per problem. In a small user study, we find a positive correlation between puzzle-solving performance and coding experience, and between the puzzle difficulty for humans and AI solvers. Therefore, further improvements on P3 could have a significant impact on many program synthesis areas.

## 1 Introduction

Puzzles are often used to teach and evaluate human programmers. Classic puzzles such as the Tower of Hanoi teach fundamental concepts such as recursion. Programming competition problems, also referred to as puzzles [34], evaluate a participant's ability to apply these concepts. Puzzles are also used to evaluate programmers in job interviews, and puzzles such as the RSA-factoring challenge test the limits of state-of-the-art algorithms. Each of these types of puzzles is described in its own format, often in a natural language such as English. Evaluations often include a hidden test set.

We introduce a novel puzzle representation called a programming puzzle or simply a *puzzle*, which captures the essence of these challenges in a form convenient for machines and programmers. At a minimum, a puzzle is specified by a function $f(y)$, and the goal is to find $y$ such that $f(y) = $ `True`. More generally, a puzzle can include input variables $x$. Then, the puzzle can be seen as an output *verifier* $f(y, x)$ that validates $y$. The answer $y$ is typically the output of a synthesized program $g$. In order to find $g(x) \to y$, a *synthesizer* is given the source code of $f$ (and possibly $x$), with the goal of generating a program $g$ such that $f(g(x), x) = $ `True`. Importantly, puzzles make for an objective and explicit programming evaluation based solely on code with no formal requirement for input/output examples, natural language descriptions, or reference solutions.

Puzzles may have multiple valid outputs $y$ and some puzzles, even if very short, are extremely challenging. Figure 1 illustrates three puzzles that are diverse in domain, difficulty, and algorithmic

---

[1] https://github.com/microsoft/PythonProgrammingPuzzles

35th Conference on Neural Information Processing Systems (NeurIPS 2021) Track on Datasets and Benchmarks.

```python
# Find a string that when reversed and concatenated with "world" gives "Hello world"
def f1(y: str):
    return y[::-1] + "world" == "Hello world"

# Tower of Hanoi, often teaches recursion. Move [i, j] means move top disk on tower i to j, with 1 ≤ i,j ≤ 3
def f2(moves: List[List[int]], num_disks=8):
    state = [1] * num_disks   # All disks start at tower 1.
    for [i, j] in moves:
        assert state.index(i) <= (state + [1, 2, 3]).index(j), "bigger disk on top"
        state[state.index(i)] = j   # Move smallest disk from tower i to tower j.
    return state == [3] * num_disks   # All disks must end on tower 3.

# Find a non-trivial integer factor d of a large number n
def f3(d: int, n=100433627766186892221372630609062766858404681029709092356097):
    return 1 < d < n and n % d == 0
```

Figure 1: Programming puzzles ranging from trivial to longstanding open algorithmic challenges in multiple domains. `f1` is solved by y=`"Hello "`[::-1], a recursive program (see Figure H.1 on page 37) outputting 255 moves solves `f2`, and `f3` requires computational number theory algorithms.

tools. The first puzzle is an easy (for humans) puzzle that tests one's understanding of basic syntax and properties of strings. The second is the quintessential example of recursion, and the last is a hard problem requiring advanced algorithms such as the quadratic sieve.

We also release a growing open-source Python Programming Puzzles dataset, called P3, which is already comprehensive in terms of difficulty, domain, and algorithmic tools. This dataset unifies many of the types of puzzles mentioned above. While P3's puzzles are all specified in Python, solution programs $g$ can be written in any language, or simulated by neural networks.

As describe in §3, P3 also contains numerous classic puzzles; optimization puzzles such as solving a linear programming; graph puzzles such as shortest path; and competitive-programming problems. The most difficult puzzles involve longstanding open problems such as learning parity with noise [11]; factoring [24, 30]; or finding a cycle in the $3n + 1$ process which would disprove the Collatz conjecture [35]. Thus, if AI were to surpass human-level performance on this dataset, it would lead to breakthroughs on major open problems. The P3 puzzles were inspired by sources such as Wikipedia, algorithms books, programming competitions, and other program synthesis datasets. This dataset is growing as an open-source project, and anyone can add a puzzle by simply writing a function `f`.

One motivation for programming puzzles is that improvements in solving puzzles may lead to performance gains at other tasks. Recently, neural Language Models (LMs) have advanced the state-of-the-art of AI systems performance in offering code completions and synthesizing source code in general-purpose programming languages, such as Python, based on English descriptions [5, 13]. While such systems represent a major advance over prior approaches, Chen et al. [13] point out that they also reproduce elementary programming mistakes. Figure 2 illustrates how state-of-the-art GitHub Copilot [63] solves a complex problem, handles the ambiguity of English, and yet makes elementary errors. Performance gains in solving programming puzzles may result in fewer errors or solving more sophisticated algorithms problems in downstream tasks such as code completion.

Puzzles are *objective*, meaning that it is easy to unambiguously evaluate whether one's own answer is valid without consulting an answer key. This evaluation also allows bootstrapping, even on test puzzles without gold solutions. Given a set of puzzles $(f_i, x_i)$, one can attempt to solve them with solutions $g_i$, determine with certainty which solutions are correct, and use those to improve one's ability to solve the remaining puzzles [19]. Inspired by success in playing games [53, 56], self-training has also proven useful in program synthesis [see, e.g., 6, 15]. Other commonly used representations, including natural language descriptions or Programming by Example (PbE), have inherent ambiguity. See Appendix F for a comparison of a competition problem represented in English and as a puzzle.

From a theoretical point of view, as we shall discuss, objectivity can be formalized as the complexity class NP of non-deterministic polynomial-time decision problems. Moreover, the puzzle decision problem is NP-complete, meaning puzzles can readily express any NP problem, including polynomial-time problems and other NP-complete problems such as Boolean satisfiability.

We compare several enumerative random forest and Transformers-based top-down solvers, as well as GPT-3 and Codex LM solvers with different prompt types (e.g., zero/few-shot and with/without English descriptions). In our experiments, without access to any reference solutions, only utilizing

```
def adjacent_primes(n: int):                          # x is the concatenation of "Hello" and "world"
    """Find the nth & (n+1)st prime numbers"""        x = "Hello" + " " + "world"
    primes = [2, 3]
    i = 3                                             assert "hello" not in "hello world"
    while len(primes) < n:
        if all(i % p != 0 for p in primes): …         assert len(filename + ".json") < len(filename)
```

Figure 2: GitHub Copilot code completion examples (in gray). Left: Copilot correctly implements a seven-line function. Top right: the completion adds a space character that may or may not have been intended by the user. Middle and bottom right: errors indicating a lack of basic understanding.

self-training bootstrapping, our enumerative models solved up to 43% more P3 problems than a naive brute force baseline. Our LM solvers were able to solve many of the puzzles, given enough tries.

To address the questions of whether puzzles measure programming proficiency and how puzzle difficulty compares between humans and computers, we performed a small user study. Puzzles were accessible and enjoyable for programmers with varying levels of experience. While both GPT-3 and enumerative techniques can solve a fraction of the puzzles, human programmers outperform them. For example, bootstrapping GPT-3 with up to 10K tries solved 60% of the puzzles, lower than both beginner and experienced participants that solved 76% and 87% puzzles on average, respectively. Overall, we find perceived puzzle difficulty to scale similarly for both humans and AI.

The main contributions of this paper are introducing:

1. programming puzzles: a new type of problem suitable for algorithmic problem-solving (for both machines and humans);
2. P3, an open-source dataset of puzzles covering diverse domains and difficulties; and
3. an evaluation of humans and baselines demonstrating that puzzles can be used to measure algorithmic problem-solving progress.

Progress in code completion is rapid—even between the time of submission of this paper and its publication, an API to Codex (a GPT-3 model fine-tuned for code completion) was released [13]. Our evaluation does in fact show significant improvements of Codex over other baselines.

## 2   Problem formulation

Programs, inputs and outputs can all be formally represented as strings, where $\Sigma^*$ is the set of finite strings over alphabet $\Sigma$. The set of verification functions is denoted by $\mathcal{F} \subseteq \Sigma^*$, with inputs and outputs $\mathcal{X}, \mathcal{Y} \subseteq \Sigma^*$, respectively. A puzzle is defined by pair $(f, x) \in \mathcal{F} \times \mathcal{X}$ and the result of running verifier $f$ on output $y \in \mathcal{Y}$ is denoted $f(y, x) \in \{0, 1\}$. Output $y \in \mathcal{Y}$ is *valid* if it *satisfies* $f(y, x) = 1$, i.e., $f$ outputs 1 when run on $(y, x)$, within a specified amount of time. To ensure that puzzles can be *quickly* verified, it is necessary to upper-bound the time required for puzzle verification. This ensures that the puzzle decision problem, namely the problem of determining whether, given $f, x$, there is $y$ such that $f(y, x) = 1$, is in the complexity class NP. Formally, the puzzle decision problem is, given strings $f$ and $x$ denoting the puzzle (represented as, say, a Turing machine) and input, and a timeout $t$, does the puzzle output 1 in time $\leq t$. See Appendix D for further details.

A **solver** takes $n$ puzzles and timeouts $(f_1, x_1, t_1), \ldots, (f_n, x_n, t_n)$, and produces outputs $y_i$ to as many puzzles as it can within a time bound $T$. Of course $T \gg \sum t_i$ is significantly larger than the verification timeouts. Formally, the *score* of solver $S : \mathcal{F}^n \to \mathcal{X}^n$ is the number of puzzles $f_i$ for which $f_i(y_i, x_i)$ outputs 1 in time $\leq t_i$. Although we do not study it in this paper, it would be natural to assign different *values* to different puzzles. For example, solving open problems such as finding a Collatz cycle or factoring the largest RSA challenge integer (currently unsolved, with a \$200,000 prize offered), should be of greater value than solving a simple hello-world puzzle.

It is convenient, though not required, to solve puzzles by outputting a program $g$ which, when run, computes output $y = g(x)$. Such a program is called a **solution** $g$. Short solutions may have long outputs, e.g., the puzzle (f=lambda y: len(y) == x, x=1000000) requires a string of length one million as solution g=lambda x: 'a' * x. In this example, $y = g(1000000)$ is a valid output of length one million. Of course, another solution would be to explicitly write a string of length one million in the code, though this implementation may not pass a human code review. In the dataset and this paper, we provide solutions since they may be significantly shorter. Many puzzles fit a single

**problem** template, meaning they share the same verifier $f$ but have different inputs $x$. Thus a dataset may have many more puzzles than problems.

## 3 The P3 dataset

P3 uses Python, the de facto language of ML research, as the programming language for specifying puzzles. At the time of publication, P3 currently has 397 problems, summarized in Table 1. The latest dataset can be generated by simply running `make_dataset.py` in the repository. More puzzles may be created by increasing the number of puzzles per problem argument, though most experiments in this paper use only one puzzle per problem. Every puzzle is described by a function with a required typed argument (i.e., the candidate output) that returns `True` upon success. Since Python is not type-safe, we add type-checking to ensure that outputs match the declared type. Figure C.1 on page 19 illustrates a puzzle where type checking is important.

We also provide code for serializing Python objects to and from strings in a json-like format, so that programs implemented in any language can produce outputs. Moreover, strings are universal in that they can encode arbitrary Python objects including functions, as in the Quine puzzle (`lambda quine: eval(quine) == quine`)[2] motivated by the classic challenge of writing a program that outputs its own source code. As evaluation of the string `quine` can lead to an infinite loop, this puzzle illustrates the necessity of the evaluation timeout $t$ for attempted solutions.

While not necessary for evaluation (since puzzles are self-contained) we follow the common practice of programming competitions and provide a reference solution to most (over 90%) of the puzzles. Some puzzles have more than one solution. A handful of puzzles represent major open problems in computer science and mathematics including Factoring (and Discrete Log), Planted Clique, Learning Parity with Noise, Graph Isomorphism, and finding a Collatz cycle,[3] as described in Appendix E. We also provide English descriptions for each puzzle in the dataset to support research involving natural language. Appendix F compares programming competition problems to puzzles.

**Creation process.** The following sources were used for identifying possible puzzles:
- Wikipedia, specifically the Logic puzzles category, the List of unsolved problems in mathematics, and the List of algorithms.
- Competitions, primarily the competitive programming website codeforces.com but also a handful of problems from the International Collegiate Programming Contest and the International Mathematical Olympiad (IMO)–a high school mathematics competition.
- Puzzles inspired by the HumanEval dataset used for evaluating Codex [13], added in v0.2.
- The Python programming language itself, with trivial puzzles created to test understanding of basic functions, such as the the hello-world puzzle which tests string concatenation.

P3 is organized topically into modules listed in Table 1. These topics include domains such as number theory, graph theory, chess puzzles, game theory, etc., as well as puzzles inspired by a specific source such as a specific programming competition. One finding in this paper is that many types of puzzles can be captured in spirit, if not exactly, as succinct puzzles. Common patterns include:
- Problems that are *naturally* puzzles. For instance, search problems such as the TowerOfHanoi (f2, Figure 1) and SlidingPuzzle simply test the sequence of moves to see if they lead to the goal state.
- Problems that have an equivalent natural puzzle. For instance, the standard definition of the factoring problem, namely factorizing an integer into its prime factors would require a puzzle that tests primality. However the simpler problem of finding any non-trivial integer factor, f3 in Figure 1, can be recursively called to solve the prime factorization problem.
- Optimization problems. Some such problems have equivalent natural puzzles, e.g., linear programming is well-known [18] to be equivalent to solving a zero-sum game which is the ZeroSum puzzle. For others, such as LongestMonotonicSubstring or ShortestPath, we specify a bound $\theta$ on the objective, and the goal is to find a feasible $y$ with objective better than $\theta$. In order to generate $\theta$ (included in $x$), we first solve the optimization problem ourselves, but the puzzle generation code is not provided to the solvers.

---

[2]GPT-3 generated a 5-character solution to the quine puzzle while the authors' solution was 88 characters.

[3]The solution to this problem would disprove the Collatz conjecture that is believed to be true, but no proof has been found yet. Therefore, if the conjecture is true, the maximum attainable score in P3 is $< 100\%$.

Table 1: Number of problems (and how many of them have at least one reference solution) per domain in P3 v0.2. The right two columns show the average size of puzzles and solutions, measured by the number of nodes in the Python AST.

| Domain | Problems | Solutions | $|f|$ | $|g|$ |
|---|---|---|---|---|
| Algebra | 4 | 4 | 70 | 172 |
| Basic | 23 | 23 | 54 | 44 |
| Chess | 5 | 3 | 221 | 186 |
| Classic puzzles | 23 | 23 | 101 | 211 |
| Codeforces | 47 | 45 | 73 | 70 |
| Compression | 2 | 2 | 126 | 113 |
| Conways Game of Life | 3 | 2 | 189 | 345 |
| Games | 7 | 7 | 225 | 299 |
| Graphs | 12 | 11 | 105 | 152 |
| HumanEval | 164 | 164 | 81 | 62 |
| ICPC | 4 | 4 | 304 | 569 |
| IMO | 6 | 6 | 173 | 256 |
| Lattices | 2 | 2 | 70 | 228 |
| Number Theory | 16 | 12 | 47 | 68 |
| Probability | 5 | 5 | 85 | 72 |
| Study | 30 | 30 | 40 | 21 |
| Trivial inverse | 39 | 38 | 27 | 30 |
| Tutorial | 5 | 5 | 27 | 13 |
| Total # / Average size | 397 | 386 | 79 | 84 |

- Problems that ask how many items in a certain set satisfy a given property, may be converted to problems that require an explicit enumeration of all such items. See for example the AllPandigital-Squares puzzle that requires all 174 pandigital perfect squares as input.
- Problems that involve game-playing can often be converted to puzzles. In chess, this includes the classic Eight Queens and Knights Tour search problems. A puzzles Mastermind involves exhibiting a winning strategy tree, and a nim puzzle involves beating a given computer opponent.

In order to ensure that each puzzle is achieving its goals, the puzzle design process has a step which automatically tests for trivial solutions such as small integers or common strings.

**Exclusions.** Many programming challenges do not make as good puzzles. First, simple *translation* tasks, where goal is translate a sequence of steps described in natural language into a program, do not make good puzzles. Second, some challenges require problem-solving that is not easily expressed as a program. For example, computing the probability of rolling a total of 350 when rolling 100 dice relies on external knowledge about probability theory. Third, "soft" challenges involving natural language or images are not in NP and not easily verifiable. This includes challenges involving human commonsense or world knowledge about names, dates, or image classification. Finally, *interactive* challenges do not make for good programming puzzles. Fortunately, several other benchmarks cover these latter two types of exclusions [see, e.g., 3, 32, 37, 42, 46, 48, 49, 51–53, 55, 58, 61, 62].

**Growth process.** The focus of this paper is in creating a framework with an initial dataset; and demonstrating its utility for developing and evaluating AI solvers. As a GitHub repository, the dataset can grow over time in a standard manner with the ability to reference previous versions. We plan to continue adding puzzles and hope that others will as well. Popular competitive-programming websites such as codeforces may be a source of thousands of puzzles of varying difficulties.

## 4 Solvers

In this section, we describe the models we develop as baselines for the dataset. We consider both solving problems independently and joint solvers that bootstrap from previously obtained solutions to find new ones. We also consider both enumerative solvers that use standard techniques from program synthesis and LM solvers that use GPT-3 and Codex to solve puzzles. While a direct comparison between these two different approaches is difficult because they run on different hardware (the LMs call an API), we can still compare the relative difficulty with which they solve different puzzles, and also to human difficulty rankings among puzzles.

### 4.1 Enumerative solvers

Following prior work [1, 6, 15, 40], we develop models to guide the search for $g$ over the space of all possible functions. In particular, we implement a grammar that generates Abstract Syntax Trees (ASTs) for a large subset of Python. The grammar covers basic Python functionality and is described in Appendix B.1. Specifically, each Python function is translated to an AST using a given set $\mathcal{R}$ of rules. Based on the puzzle, a context-specific distribution over rule probabilities is computed. To facilitate efficient top-down search, the context of a rule is defined to be the rule used by the parent node and the index of the current node among the parent's children. Thus if the parent node was a division binary operator, then the two children would each have different contexts, but if two such divisions were present in the same program, both numerators would share the same context.

Each puzzle $f$ is represented by a feature vector $\phi(f)$ and each context is represented by a vector $c(p, i)$ where $p$ is the parent rule and $i$ is the child index. Each rule $r \in \mathcal{R}$ is also associated with a feature vector $\rho(r)$. The probability distribution over $\mathcal{R}$ is determined based on $\rho(r), \phi(f), c(p, i)$, and the likelihood of a solution $g$ is the product of all rules constituting its AST. Naturally, this scoring mechanism introduces a bias towards shorter programs (i.e., smaller trees), which is desirable as a short solution is easy to inspect.

`COPY` **rules.** Solutions often reuse constants or puzzle parameters, for example the constant 25 or the variable `s` in example `f2` in Figure 1. As in prior work [40], for each puzzle, the global rules bank is expanded to include `COPY` rules for constants and parameters of the examined puzzle.[4] When composing solutions, this rule can reduce the complexity of the solution by simply learning to copy part of the puzzle rather than having to generate it from scratch. For simplicity, we create copy rules for each of the supported types and assign the probabilities uniformly across all the puzzle's constants of that type. In other words, our models learn when a certain type should be copied from the puzzle, and rank all available constants and parameters of that type the same.

To solve a new puzzle, we perform a top-down search. Specifically, at each node, we apply a selected model over all rules in $\mathcal{R}$ whose type matches the context, and re-normalize the scores to create a valid probability distribution. The solver enumerates solutions in order of decreasing likelihood until it finds a solution $g$ such that `f(g())` evaluates to `True` in time $\leq t$, for a maximum number of tries $M$. See Appendix B for details on the search and rules. Next, we briefly describe our models.

**Uniform.** The first model is a simple uniform rule that assigns the same probability to all rules. The only exception is `COPY` rules, which have a larger, fixed probability in order to bias the solver towards utilizing this option. As we score programs by their joint probability, this bias effectively favors shorter programs. We use this model to find solutions to the easier problems, satisfied by a simple and short answer, and use these to bootstrap the learning of the parametric models. This model also provides a naive brute force baseline to compare the parametric models with.

The remaining two models have parameters that are fit based on *bootstrapping*. Namely, given previously obtained solutions, we collect all parent-child rule pairs as self-supervision and fit the model's parameters on them. The training size is then the total number of nodes in all the trees among solutions discovered up until that point. We implement two bigram parametric models to predict $\mathbb{P}\big(r \mid \rho(r), \phi(f), c(p, i)\big)$, where $r$ is a candidate rule to appear in $g$'s tree under $p$ as its $i$'s argument.

**Random forest.** In this model, we represent $f$ as a bag-of-rules $\cup\{r_k \in f\}$. Specifically, $\phi(f)$ is a vector of length $|\mathcal{R}|$ representing the number of occurrences of each rule in $f$. $p$ and $i$ are encoded as a one-hot vector and concatenated to $f$'s representation to construct the input to the model. Given past solution trees, we train the model to predict the index of $r$ out of $|\mathcal{R}|$ given $f, p, i$ examples.

**Transformer.** Following the recent success of transformer models [20, 57] in encoding source code [21, 31, 54, *inter alia*], we turn to these encoders for richer representations. We use a RoBERTa-based [36] Transformer to encode puzzles and rules directly from their code. The probability of a rule $r$ being the $i$'s child of $p$ in $g$ is proportional to the dot product of the deep joint representation of $f, p, i$ and the Transformer encoding $\rho(r)$. We pretrain the Transformer with a masked language model task on Python GitHub repositories [27].[5] Then, our solver concatenates the Transformer

---

[4]When executing a solution, `COPY` rules are simply the identity function (`COPY = lambda x: x` in Python).

[5]Our pretrained model and tokenizer are available at `https://huggingface.co/tals/roberta_python`.

```
def f(li: List[int]):
    return len(li) == 10 and li.count(li[3]) == 2

assert True == f(...
```

Figure 3: A Short prompt for a puzzle requesting a list of ten integers where the fourth item occurs exactly twice, with valid completion ...[1,2,3,4,5]*2). Appendix C has Medium/Long prompts.

encodings $\phi(f)$ and $\rho(p)$ with a learned embedding for $i$, following by non-linear layers to compute the joint representation. We fine-tune the solver on parent-child rule pairs from previously acquired solutions. See Appendix B.2 for extended details, and Figure B.1 on page 19 for a model diagram.

## 4.2 Autoregressive Language Model solvers

We experiment with the transfomer-based GPT-3 [12] and Codex [13] LMs with billions of parameters. Codex was trained over large amounts of publicly available code from GitHub, specifically aimed for coding applications. We follow the recent strategy of designing a prompt that directs the text generation to our desired task. This approach has shown to be useful in converting natural language descriptions to programming code and guide theorem proving [45]. Unlike our enumerative models that build an AST, LMs generate the solution as a string that is directly evaluated as Python code.

We consider four different prompts: (a) A *short* zero-shot prompt based solely on the puzzle at hand (illustrated in Figure 3); (b) a *medium* 5-shot prompt that includes the five example puzzles that had been shown to (human) programmers during our user study (Appendix Figures C.2-C.3); (c) a *long* prompt with the same five puzzles augmented by English descriptions of the tasks in comments (Figures C.4-C.5); and (d) a *bootstrapping* prompt which uses only solutions to problems that it has already solved (Figures C.7). The bootstrapping prompt begins with no solutions but quickly exceeds the API maximum length as more puzzles are solved. At that point, previously solved puzzles are randomly sampled to form the prompt. The prompts used for Codex are slightly more sophisticated but enable multi-line programs.

The completions which parse as valid Python expressions are then evaluated. Appendix C gives further details of the execution environment, the API parameters and other prompts we investigated.

## 5 Experiments

We use our P3 dataset to evaluate the performance of the solvers from §4. We assume no access to reference solutions[6] and measure how many puzzles are solved by each solver with up to $k$ tries per puzzle, where each try is a potential solution that is evaluated. For the enumerative solvers, this is equivalent to having a valid solution ranked in the top $k$. For LM solvers, we use **pass**@$k$ [13] which is an unbiased estimator of the probability of obtaining a solution within $k$ tries. First, we test the solvers bootstrapping efficacy in leveraging previously solved problems to solve new ones. Then, once solutions to a single instance of certain problems are found, we test whether solvers also succeed on other problem instances (i.e., puzzles originated from the same problem). In §5.1, we present our user study results that compares human's performance with AI solvers. Finally, in §5.2, we test whether P3 can distinguish between subtly different variants of Codex, using the larger v0.2 release of the dataset (the current version at the time of publication).

**Learning from past solutions.** This first experiment was run on the v0.1 release of P3.[7] We use a single puzzle instance per problem. We first identified the 138 of the 200 v0.1 problems supported by our grammar (see Appendix B.1). For the enumerative solvers, we then ran the uniform solver with $k = 10^4$ on these 138 problems supported by our, solving 38 of them. The solutions contain a total of 2,475 rules that we use to train the parametric models. In the bootstrapping variant, we repeat the training for 6 cycles, each time adding the new solutions found with $k = 10^4$. In the final round, we allow up to $k = 10^6$ solving tries (including previous cycles). For comparison to GPT-3/Codex,[8] we

---

[6] With the exception of the Medium and Long prompts that including five Tutorial problems and solutions.

[7] https://github.com/microsoft/PythonProgrammingPuzzles/tree/v0.1

[8] The Codex API was released after this experiment had been run on v0.1 using the enumerative and GPT-3 solvers. Thus, we simply replaced the GPT-3 solver with the Codex solver and re-ran on the same 138 puzzles.

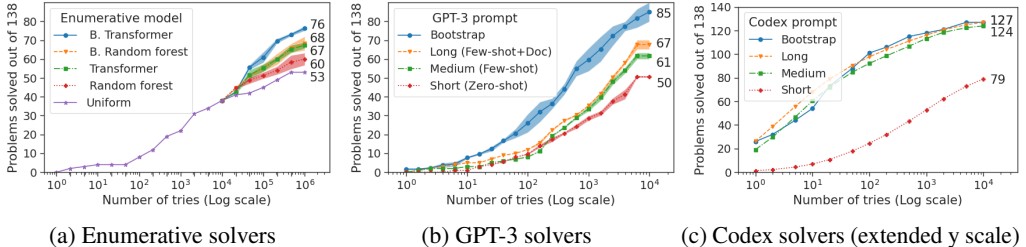

|     |     |     |
| :-: | :-: | :-: |
| (a) Enumerative solvers | (b) GPT-3 solvers | (c) Codex solvers (extended y scale) |

Figure 4: Increasing the number of tries allows solving new problems. Better solvers, though, solve new problems significantly faster by learning from past experience. Parametric enumerative solvers (a) initialized with the solutions of the uniform solver at $k = 10^4$ accelerate the solution search. Additional self-training bootstrapping cycles (marked with B.) solve even more problems. GPT-3 (b) and Codex Davinci (c) solvers were evaluated with up to $10^4$ attempts. Having natural language descriptions (Long) provides small improvements over Medium. Adding previously found solutions to the prompt (Bootstrap) allows significant improvements for enumerative and GPT-3, and matches Long for Codex. Overall, the Codex models performed best, solving up to 127 of the examined 138 puzzles. (a), (b) are averaged across three runs and the shaded areas show the standard deviation.

use the same 138 problems and start with a zero-shot prompt. As valid solutions are found, they are appended to the prompt as discussed in §C.

Figure 4a shows the total number of puzzles solved by each enumerative solver, with and without the self-training bootstrapping cycles. We report the average results across three runs and present the standard deviation in the graph. We see that the parametric models quickly improve over the naive uniform search and that the bootstrapping process facilitates solving many new problems. At $k = 10^6$, the random forest and Transformer-based enumerative models solved a total of 68 and 76 problems, respectively, which is 28% and 43% more than the uniform solver.

The GPT-3 solver also improves by learning from previously found solutions. As Figure 4b shows, few-shot settings with tutorial examples perform better than zero-shot (Short) and solve new problems. Including natural language descriptions (Long) helps for solving five more puzzles, with up to $10^4$ tries. The best strategy, however, is the bootstrapping one that starts without any reference and adds solutions to the prompt as they are found. Codex, trained on large amounts of code, performs the best (see Figure 4c) but does not benefit significantly from bootstrapping.

**Generalizing to other problem instances.** In the previous experiment, we attempted to solve the *default* single puzzle instance of each problem. Next, we examine whether our solvers can also solve other puzzle instances, originating from the same problems. We collect a set of 700 puzzles that are random instances of 35 problems for which both our bootstrapping enumerative models solved the default puzzle. At $k = 10^4$, the random forest and Transformer models solved 75% and 79%, respectively. As a reference, the uniform model solves only 62% of these puzzles.

### 5.1 User study

In a small user study, 21 participants with varying experience in Python programming attempted to solve 30 puzzles, as found in v0.1 dataset as the `study` module. Each puzzle was allocated a maximum of 6 minutes to solve, and the study was conducted virtually using Jupyter notebooks. Participants were employees at a major software company and were recruited by email and at a hackathon. No compensation was offered. Participants were first given a short tutorial about puzzles and how to submit solutions. The user study files are available in the open-source dataset, and Appendix G has further details including the 30 puzzles.

The first finding is that success in puzzles correlates with programming experience. For our retrospective study analysis, we split the participants by the median years of Python programming experience. We had 10 *beginners* with less than three years of experience, and 11 *experienced* participants with at least three years. We find that 9 of the 30 puzzles were solved by all beginners, while 17 of the puzzles were solved by all experienced participants. Also, beginners spent on average 194 seconds per puzzle, while experienced spent only 149 seconds on average. The average solving time provides

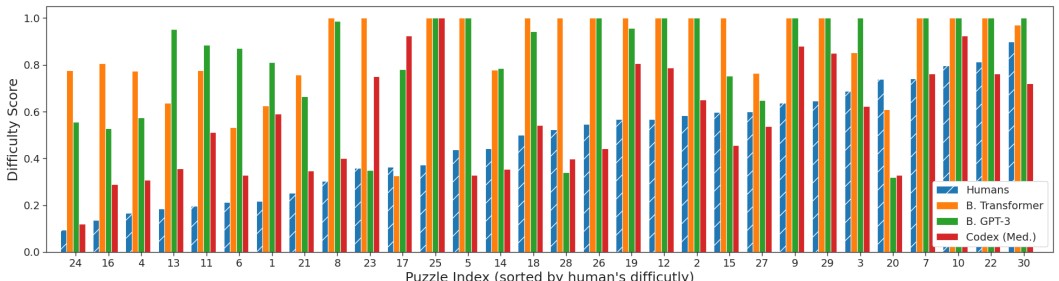

Figure 6: The difficulty score per study puzzle for both humans and AI solvers, sorted by the human's scores. The difficulty score for humans is measured by the average fraction of solving time out of the maximum allowed. For AI, we use the fraction of allotted attempts required. Most of the puzzles solved by AI (low difficulty score) are also easier for humans (left hand side of the plot).

a useful proxy to the perceived difficulty of each puzzle. Overall, we see that puzzles are easier for experienced programmers, indicating their value for evaluating programming proficiency.

Next, we compare human's performance to Codex-davinci. We use the Medium prompt as it is similar to the study format (i.e., same 5 tutorial examples, no docstrings). Participants solved an average of 24.6 out of the 30 puzzles (22.9 for beginners and 26.2 for experienced) within the 6 minutes per puzzle time limit. Only one out of the 21 participants solved all puzzles. As Figure 5 shows, Codex required 1K tries per puzzle to match the performance of beginner programmers in our study.

Finally, we find that difficult puzzles for humans are also harder for AI. Figure 6 shows that most of the puzzles solved by AI solvers with limited number of tries are the ones that are easier for humans (i.e., solved faster). To compare the two, we define a puzzle's perceived difficulty score as the average solving time for humans and the expected number of required tries for

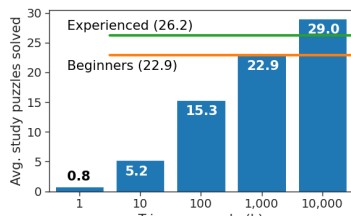

Figure 5: Number of solved puzzles by Codex-davinci (blue bars), compared to human coders with 6 minutes per puzzle (horizontal lines).

machines (normalized to $[0, 1]$, where the score of unsolved puzzles is set to 1). The Spearman's rank coefficient of humans with B. GPT-3 is 0.512, and with Codex (Med.) is 0.563. The AI solvers correlation is stronger with beginner programmers (0.541 and 0.562), than with the experienced ones (0.470 and 0.544, respectively). On the one hand, this suggests that additional computational power might allow AI solvers to match humans. However, as Figure 4 shows, this improvement is logarithmic, leading to diminishing returns. Encouragingly, we see that even within the same budget, modeling choices can improve performance. We hope that P3 will support the research and development of new AI solvers that will solve more puzzles with less computational effort.

## 5.2 Comparing small and large Codex models

In addition to the standard *davinci-codex* engine, the API offers an alternate *cushman-codex* engine that they report is significantly faster and only slightly less accurate. To test the ability of P3 as an evaluation of such fine distinctions, we ran the Medium and Long prompts on both engines across the most recent v0.2 release[9] of 397 puzzles. As can be seen in the results of Table 2, the larger engine indeed slightly outperformed the smaller

Table 2: Codex **pass**@$k$ results over P3 v0.2.

| engine (prompt) | $k = 1$ | 10 | 100 | 1,000 |
|---|---|---|---|---|
| cushman (Med.) | 7.1% | 26.7% | 51.7% | 68.3% |
| davinci (Med.) | 11.2% | 36.7% | 60.6% | 75.3% |
| cushman (Long) | 14.9% | 42.4% | 63.9% | 76.5% |
| davinci (Long) | 18.3% | 48.7% | 69.1% | 79.8% |

engine across all $k$. Thus, in this experiment, puzzle solving success aligns with code completion success. Also, we observe that English descriptions (Long prompt) are helpful for both engines. Inasmuch as puzzles are useful for code completion, the $< 20\%$ success rates at $k = 1$ leaves substantial room for improvement.

[9] https://github.com/microsoft/PythonProgrammingPuzzles/tree/v0.2

Table 3: Codex-davinci and Codex-cushman number of solved problems per domain with up to 1,000 tries for Medium and Long prompts. The first row also shows the number of available P3 v0.2 problems in that domain. The score is the average percent solved across domains.

| Model | Algebra | Basic | Chess | Classic | CodeForces | Compression | Conway's | Games | Graphs |
|---|---|---|---|---|---|---|---|---|---|
| cushman (Med.) | 3/4 | 15/23 | 0/5 | 6/23 | 32/47 | 0/2 | 0/3 | 1/7 | 6/12 |
| davinci (Med.) | 2 | 20 | 0 | 8 | 35 | 0 | 0 | 1 | 9 |
| cushman (Long) | 2 | 21 | 0 | 5 | 38 | 0 | 0 | 1 | 8 |
| davinci (Long) | 2 | 22 | 1 | 4 | 39 | 0 | 0 | 1 | 8 |

| Model | HumanEval | ICPC | IMO | Lattices | N. Theory | Probability | Study | Trivial$^{-1}$ | Tutorial | Score |
|---|---|---|---|---|---|---|---|---|---|---|
| cushman (Med.) | 139/164 | 2/4 | 1/6 | 0/2 | 8/16 | 2/5 | 21/30 | 33/39 | 5/5 | 44.2 |
| davinci (Med.) | 145 | 2 | 1 | 1 | 9 | 3 | 22 | 36 | 5 | 51.2 |
| cushman (Long) | 149 | 1 | 1 | 1 | 9 | 3 | 24 | 36 | 5 | 49.8 |
| davinci (Long) | 155 | 1 | 1 | 2 | 10 | 3 | 25 | 38 | 5 | 54.8 |

Table 3 shows the number of achieved solutions per domain, as well as an overall score computed as the macro-average of solving rates across domains.

# 6   Related Work

Program synthesis has taken drastically different forms for different applications, often resulting in one-off evaluations rather than common datasets. A major paradigm is Programming by Example (PbE) where problems are specified by input-output examples. For instance, several studies focus on text processing [22] or robot navigation [43]. While convenient for end user applications (e.g., many in [44]), PbE alone is inadequate to objectively describe many sophisticated algorithmic programming challenges. A recent ARC dataset [14] adopts PbE for evaluating abstraction and reasoning in AI, but as in all PbE applications, there can be ambiguity.

Program synthesis from formal specifications has a long history of study [surveyed in 23], benchmarked by e.g., the SyGuS competition [2]. In this setting, however, the AI system has to synthesize an algorithm that correctly and efficiently solves a problem on all inputs (and often prove correctness as well). Writing and testing such formal specifications is often non-trivial.

English descriptions, often mixed with examples, are becoming an increasingly popular problem representation as LMs improve [28, 33, 60]. In independent work, Hendrycks et al. [26] created a large dataset of English programming problems with examples on which they fine-tuned GPT models. In another concurrent work, the Codex model that powers the new GitHub Copilot auto-completion tool [13] was evaluated with short problem descriptions paired with a set of unit tests that should validate the described specification. Our work, together with this very recent and concurrent work [5, 13, 26], represent the first controlled evaluation of large Transformer-based LMs on general-purpose program synthesis.

The recent CodeXGLUE benchmark [38] collected several code-related datasets. To evaluate generation, they use CodeBLEU [47] which relies on ASTs and other code-specific aspects. This evaluation still requires reference solutions and, therefore, does not resolve the answer-key bias with ambiguous specifications. Several neighboring fields that have made substantial progress in reasoning include theorem proving [8], two-player game playing [53], and SAT-solving [9]. In all these fields, important progress has been made by encoding the problems, be they theorems, game rules, or optimization problems, in machine-readable formats that do not involve the ambiguities of natural language.

# 7   Conclusions

We introduce Python Programming Puzzles (P3), an open-source dataset with puzzles described only in source code. As discussed in §3, the puzzle framework captures NP problems, which include a wide range of interesting challenges. Puzzles allow fast and objective evaluation, thereby supporting unsupervised solving without training solutions. We implemented and evaluated several enumerative program-synthesis and LM baselines, and found a positive correlation between their per-puzzle performance and the difficulty for human programmers. Similarly, LMs that performed better at code completion also solved more puzzles with less tries.

We welcome contributions to P3 and hope it will grow in size, coverage, and utility.

**Acknowledgments.** We would like to thank Mariia Mykhailova for suggesting doing a Python Programming Puzzles Hackathon. We are especially grateful to the participants in our user study and hackathon. We are grateful to the creators of Codex and GPT-3 and to Nicoló Fusi for suggesting its use in this project. We would like to thank David Alvarez Melis and Alec Helbing for suggesting quine puzzles. We are grateful to Ana-Roxana Pop for helpful discussions and feedback. We also thank Tianxiao Shen, Adam Fisch and the rest of the MIT NLP members for valuable writing feedback.

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
