# OpenReview forum: "Programming Puzzles"
_NeurIPS.cc/2021/Track/Datasets_and_Benchmarks/Round1 — NeurIPS 2021 Datasets and Benchmarks Track (Round 1)_

### Official Review · Reviewer_XNU2 · 2021-07-01
**Interesting and impactful paper**

**Rating:** 8
**Confidence:** 4
**Correctness:** The dataset is constructed in a sound…
**Clarity:** The paper is very well-written and th…

**Strengths:**

Update: after the discussion period, I keep my rating at a clear accept. I thank the authors for responding to the reviewer suggestions.


This paper is interesting and is a very novel type of dataset. Since this dataset is novel and relevant to problems AI programming and possibly AGI, it has the potential to have a high impact. This dataset can serve as a benchmark for teaching AI to evaluate/teach python code.

The dataset has impressive breadth, and the fact that answers are verifiable with python programs is very clean.

This dataset also has the potential to benchmark progress in creating AI that can solve complex IMO math problems and unsolved math conjectures.

The paper is clear and well-written.

**Weaknesses:**

(1) I didn’t see too much discussion on the explicit purpose of this benchmark. I noticed in the datasheet, the authors say “The dataset is created to evaluate and possibly teach machines to program in Python”, and this is also briefly mentioned in the discussion. A more detailed description would improve the paper. Since this benchmark has a very broad set of complex challenges, I am curious about the applications of this dataset in progressing research in artificial general intelligence (AGI). It seems similar to the ARC challenge by Chollet, but using python instead of images. Any comment from the authors would be appreciated.

(2) The dataset should be modified before it is ready for fair research, in a few different ways.

This dataset has 139,072 total points, but only 200 problem types, and the number of puzzles per puzzle type is highly variable (e.g., 10,762 number theory puzzles and 2000 game theory puzzles). Before this dataset is used in research, I recommend making a version that weights each problem *type* equally. E.g., currently solvers that achieve top performance would be biased toward number theory over game theory. This can be done either by assigning scores to the problems as the authors mentioned (but not just by perceived difficulty - also to weight problem type equally), or by only including exactly 2 or 3 problems per type of problem.

I recommend proposing a train/test split. And it seems important to make sure that the test set contains a large amount of problem types that are not in the training set, so that solvers must learn to solve new types of problems.

Give clear version numbers, along with python version and any other specific versions. (If any of the puzzles need other python libraries, that will be important as well.)  Although it is nice that this dataset only needs to execute code to create the labels, it is important that this code is executed in the exact same way e.g. when comparing solvers from different groups of researchers. This is different from most datasets that merely have labeled test sets.

Finally, does P3 have commented/annotated and un-annotated versions? It will be important to mention this detail e.g. when evaluating GPT-3 type solutions.

(3) The authors did not discuss potential negative aspects of the work, checklist 1.c. I don’t see potential ethical issues, but it is still good practice to discuss this.

(4) Not a weakness, and more of a nice-to-have: it would be nice to see this as a contest one day, similar again to the ARC challenge.

**Additional Feedback:**

Grammatical errors and minor comments
- Unfortunately there are a large number of broken hyperlinks throughout the paper. It seems that this occurs any time there is a link to programming puzzles on Github. Example: Line 108: the link to the Collatz problem is broken. All of these should be fixed.
- Figure 1, in the return statement of f3, it should be “1 < d < n” ? Otherwise, inputting 1 would return True.
- The Collatz Conjecture is used as an example many times throughout the paper. But it is not the best example because most mathematicians believe it to be true (source: Wikipedia), so it is unlikely that any program will ever find a counterexample to that programming puzzle. (As a consequence, it is likely impossible to achieve a perfect score on the P3 dataset.) You should mention this at least as a footnote.
- Line 46. “were to surpasses” -> “were to surpass”
- Line 52. “predict the next.” -> “predict the next one.”
- Typo on line 336. “problems and out puzzles”
- Line 167, Aren’t there primality tests which do not give prime factors?
- Line 171, give some citations here.
- Line 174, typo: “W plan”
- Table 1, typo: “Conways” -> “Conway’s”

**Documentation:**

The GitHub repository has sufficient detail on the data, notebooks, demos, contributing, and guidelines. They also fill out a datasheet which is appreciated.

Will the code to reproduce the experiments be released publicly? I encourage the authors to do so.

**Ethics:**

There are no ethical concerns that warrant further review.

**Relation To Prior Work:**

Related work is sufficiently discussed in Section 6. Although, see weakness (1).

**Summary And Contributions:**

The authors introduce a dataset for python programming puzzles. These puzzles cover a large variety of challenges, from easy string manipulation, to college-level coding questions, to competition math and computer science problems, as well as unsolved conjectures. Verifying a solution only requires running a python program. The authors also run baseline experiments with GPT-3, and give a study evaluating the human-level and AI-level difficulty of the puzzles.

---

> ### Author Response · Authors · 2021-07-13
> **That you for your comments**
>
> Thank you for the helpful comments. Our response to individual comments is provided below. We will upload an updated version of the manuscript that addressed these points soon.
>
> * **discussion on the explicit purpose and relation to ARC challenge**: We will clarify this point in the updated manuscript and discuss the relation to the ARC challenge. Indeed, one way to look at P3 is as a code-based analog of the ARC challenge.  We believe our puzzles can provide objective evaluation for the ability of AI models to solve complex algorithmic problems that require predicting the outcome of code execution and composing advanced algorithms in code. This could be valuable when evaluating new AI models (see for example GPT-3 and the new Copilot/Codex model).
>
> * **Weighting domains equally**: Thank you for this suggestion. The domains in the current dataset version vary in their difficulty and, therefore, we believe that it is informative to examine the specific performance on different domains as we report in Table 2. Yet, we agree that a combined normalized score is useful, and (following this comment) we will include a macro average of solving rates across domains.
>
> * **train/test split**: Our current setting is completely unsupervised (except for the GPT-3 medium and long prompts that use 5 tutorial solutions). This evaluation setting assumes a list of problems and the AI model should attempt to solve as many problems as possible with a minimum number of attempts. This unique setting enables test-time bootstrapping where models can leverage previous solutions to learn to solve new problems, without concerns of overfitting because no solutions were used by the learner. However, we agree with you that other supervised settings could also be interesting for future research and it will be interesting to examine in-domain and cross-domain generalization from reference solutions. In the future, in larger versions of the dataset, we will include an explicit train/test split. Meanwhile, we invite other researchers to explore utilizing the P3 dataset in the way that best suits their purpose.
>
> * **Python version and external libraries for reproducibility**: This is a good point and we will include an explicit reference version of CPython*. None of our current puzzles require any libraries that are not already included in the Python standard library. At the moment we plan to keep this constraint in order to maintain the generality of the evaluation and not to require AI models (or humans) to learn any libraries beyond basic Python. Our framework, however, readily supports problems using other libraries and in a future version we will consider including them (e.g. numpy). As a side note, in our user study, some participants reported that the puzzles helped them learn new python functionalities (e.g., “all”). Therefore, it will be interesting to explore using the puzzles format to teach both humans and AI new libraries.
>
> * **Version with/without docstrings**: Our current dataset generation script includes the problem docstrings in the markdown file but not in the json files. We will update the script to allow including them in the json files.
>
> * **Potential negative aspects**: We also did not see any obvious potential negative societal impacts of our work beyond those that might arise with advancing the state of the art in program synthesis (e.g., energy consumption, potential loss of jobs, abuse by evil parties).
>
> * **Creating a contest**: We agree with this comment. As we continue enlarging our dataset, we will consider organizing such a contest.
>
> * **Broken links and typos**: Thank you for the comments. We will fix the typos and add a footnote about the conjecture. The links to the Github repository seem to work for us. For example the collatz link in line 108 takes us to [this link](https://github.com/microsoft/PythonProgrammingPuzzles/blob/main/problems/README.md#collatzcycleunsolved) (we are using Chrome). Are you still experiencing broken links?

---

> > ### Comment · Reviewer_XNU2 · 2021-07-13
> > **Reply**
> >
> > Thanks for the reply. I agree with the authors' comments.
> >
> > I agree that since most follow-up work will likely be fully unsupervised, a train/test split is not necessary.
> >
> > I thank the authors for adding a macro average to Table 2. This seems especially beneficial for future work that uses the dataset.
> >
> > Regarding the broken links, I reviewed the paper by downloading the pdf and then opening it in Preview. It turns the links into e.g.:
> > https://github.com/microsoft/PythonProgrammingPuzzles/blob/main/problems/README.md%23collatzcycleunsolved
> > So it looks like this is not your fault. It is a problem with hyperlinks containing a "#" with hyperref + Preview, but as you pointed out, the link is correct when the paper is viewed in browser (https://openreview.net/pdf?id=fe_hCc4RBrg).
> > I’m not sure if it’s possible to have a solution work in both settings.

---

> > > ### Author Response · Authors · 2021-07-13
> > > **Some readers on Mac don't support links with hash sign**
> > >
> > > Thank you for pointing out the issue with the links and helping us debug it.
> > >
> > > I just tried to read it with preview (version 10.1) and it worked. Apparently, there is a bug with some pdf viewers on Mac: https://tex.stackexchange.com/questions/555559/href-broken-links-due-to-url-encoding-hash-sign-23.
> > >
> > > I think we can probably leave it like this and hope that not too many users will experience this bug. Other alternatives could be to avoid any location-specific links, or to create short URLs for all of them and link to them.

---

### Official Review · Reviewer_whud · 2021-07-04
**Varied problems for ML coding fit into easy-to-evaluate and straightforward form**

**Rating:** 8
**Confidence:** 4
**Clarity:** The paper is well written and easy to…

**Strengths:**

1. **Task novelty.** The introduced formulation of automatic problem-solving as a programming puzzle is to some extend novel. At the same time, it seems to capture the desired characteristic of datasets for program understanding and synthesis. The problem is formalized, well-described, and presented with related works from the field.
2. **Dataset quality.** Creation of the dataset was non-trivial and involved manual preparation of problem templates by the paper's authors. Problems seem varied and have meaningful taxonomy applied, allowing one to diagnose model concerning other solutions and human baseline, category-by-category.
3. **Problem selection.** The selection of problems seems well-conceived, e.g.,  there is a substantial gap between systems and human performance, and puzzles represent a broad spectrum of disciplines. Problems are demanding even for humans as junior/mid programmers underperform compared to developers with broad experience. At the same time, even the most experienced struggled in completing the task within the given time limit.
4. **Good, reproducible baselines.**  Evaluated models are good enough to obtain meaningful insights on the problem, and they are well described in the paper and appendix. The shared source code (declared to be opened after obtaining internal approval) seems to make the results fully reproducible after spending some time on confronting it with paper (see Weaknesses).

**Weaknesses:**

I found no major flaws. Nevertheless:

1. **Single run.** Error bars are reported only for human annotation. The authors state that solvers were not run more than once due to computational limits, but this is not conceiving and would be an argument against the dataset if true. It has been shown that different random initialization or data ordering can result in considerably higher scores (e.g., Dodge et al., 2020), and, consequently, future reviewers would demand multiple runs from solvers whose scores will be close to the current state of the art (especially, for trainable solvers relying on previously obtained solutions). If the team consisting of people from MIT, Allen, and Microsoft Research cannot afford multiple runs, who can and how can we ensure reliable comparison in the future?
2. **Source.** The accompanying source code would benefit from better README files and a clean-up. In particular, it is not clear which commands to run while trying to reproduce results and whether the commented-out commands were used. As baselines and source code are of critical value at the NeurIPS Datasets&Benchmark track, I expect a minimalistic tutorial on running the models.
3. **Varia.** There are a few statements not related to the paper essence that I find to be incorrect.
    - The described "BERT-based Transformer" seems to be the RoBERTa. Although the README file describes introduced changes in tokenization, it is still a byte-level BPE not present in the original BERT. Moreover, as far I understand, you started from the English roberta-base checkpoint. All things considered, the RoBERTa authors deserve credit here.
    - The statement regarding a "typical IQ test" is controversial and not supported by any reference or justification. If the authors had in mind the Raven's Progressive Matrices test, then the common notion is that it has only one justifiable answer. If the authors insist it is not true, such a view requires support from relevant psychometric works or broader discussion. As it seems to me there is no gain from mentioning IQ tests here, especially for their criticism, I would skip this part in the camera-ready version.


**Additional Feedback:**

Some readers may be interested in meta-information, like how much time was required to prepare a single problem to include in the dataset. It might be useful when planning similar efforts in the future.

Despite the mentioned weaknesses (am I missing something there?) I enjoyed reading the paper and consider it a precious contribution.

[Updated July 20 2021] Thanks to the authors for their engagement with the reviews, and introduced changes.

**Correctness:**

The dataset was constructed in a sound and verifiable way. Though the complete one-by-one list of considered problems nor information on why the concrete one was (not) included is unavailable, the authors provided a list of the exclusion criteria and common reasons for the problem to be disregarded. It seems to be enough in this case. The paper and accompanying appendices describe the preparation process in detail. Experiments are sound, apart from the fact that models guiding the search were trained only once.

**Documentation:**

As described in Correctness, Summary, and Strengths, I found the dataset's creation and its content sufficiently documented. Intended uses are clear in spide of the paper's content and were described in the accompanying datasheet. The licensing is clear, whereas hosting and maintenance is self-explanatory, with Github chosen as the distribution platform. Nevertheless, it was described in the datasheet as well. I don't think this paper would benefit from ethical considerations.

**Ethics:**

Nothing to report.

**Relation To Prior Work:**

There is a concise description of related work. Additionally, differences between the proposed puzzle representation and the relation of included problems to others, including competitive-programming problems, are stated implicitly or explicitly in several places of the paper and appendices.

**Summary And Contributions:**

The paper presents a concept of programming puzzles to evaluate methods for program understanding and synthesis, introduce an open-source P3 dataset, and evaluate human programmers, prompted language model, and diverse enumerative solvers.

Puzzles vary in terms of difficulty, from introductory programs written by beginners to open Computer Science problems whose prospective solutions would lead to winning the associated prize. The authors introduced 200 problems consisting of from five to 32k puzzles each. Though this number is sufficient for performing experiments, it is assumed to be an initial contribution – the constantly growing base of problems will be maintained as a Github repository, making it possible to improve the P3 dataset over time. The creation process and considered problem sources were described in detail within the paper. Moreover, it is accompanied by a datasheet covering details of its construction and composition.

The conducted experiments and proposed baselines were used to answer the questions such as how well the dataset measures programming proficiency, how puzzle difficulty compares between humans and computers, or how models solving problems independently perform compared to solvers bootstrapping from previously obtained solutions.

---

> ### Author Response · Authors · 2021-07-13
> **That you for your comments**
>
> Thank you for the helpful comments.  Our response to individual comments is provided below. We will upload an updated version of the manuscript that addressed these points soon.
>
> * **Single run**: We did not get to run the additional experiments by the submission deadline but we completed additional runs of the enumerative solvers and will add error bars to an updated version of the paper. We are currently running two additional runs of the GPT-3 benchmark.
>
> * **Sharing source code of baselines**: We are in the process of releasing the code to recreate the baselines as part of the repository, together with instructions. We understood the importance of this from multiple reviews and we expect to share it soon.
>
> * **Clarify the use of RoBERTa**: We agree with your point. We mentioned this in the appendix but we will modify the Transformer paragraph in section 4.1 to clearly state that we base our model on RoBERTa. Indeed, We initialize the pretraining on Python code with the publicly released model that was pretrained on English. However, we retrain the BPE tokenizer on the Python data and share embeddings for mutual tokens and randomly initialize the embeddings for the new tokens.
>
> * **typical IQ test**: We will remove our remark in lines 51-52 about IQ tests.
>
> * **Meta-information (e.g. time to compose a new problem)**: Some easy codeforces problems were translated in ~10 minutes each (including time to encode solution and generate tests), while some of the more challenging puzzles such as the sliding puzzles, mastermind, and several ICPC/IMO problems took several hours apiece to create. The effort is similar but slightly easier than designing the corresponding homework exercises in a programming course, because the English descriptions do not need to be checked for clarity--writing and debugging the puzzle and solution suffice.

---

> ### Author Response · Authors · 2021-07-14
> **Updated paper and code for solvers**
>
> Thank you again for your helpful comments.
> We have updated the manuscript and added code for reproducing our main experiments to our repository: https://github.com/microsoft/PythonProgrammingPuzzles/tree/main/solvers
>
> Please let us know if you have any other comments.

---

### Official Review · Reviewer_J7rM · 2021-07-04
**A comprehensive and well-documented dataset with insightful benchmarks.**

**Rating:** 8
**Confidence:** 3

**Strengths:**

1. The "puzzle" representation is powerful and precise. It can encode a wide range of programs while allowing objective evaluation and template-based instance generation. It can also be extended to include English descriptions by adding comments in the code.
2. The dataset is well documented. Each puzzle has an English description. Most puzzles have a reference solution. There are also notebooks on how to use the APIs. The framework is also very extensible, allowing the contributions of new puzzles.
3. The provided baselines reflect the current state-of-the-art. They are not trivial baselines. They can be very helpful for future research.

**Weaknesses:**

1. All models do not solve any questions from hard domains like Algebra, ICPC, and IMO. Maybe including some easy examples in these domains can help to bootstrap the models.
2. No human baseline for the whole dataset.

**Additional Feedback:**

Mention or compare against two recent related works:
CodeXGLUE: A Machine Learning Benchmark Dataset for Code Understanding and Generation (https://arxiv.org/pdf/2102.04664.pdf)
GitHub Copilot: Your AI pair programmer (https://copilot.github.com/)



**Clarity:**

The paper is well written with plenty of examples that are very helpful for understanding.

**Correctness:**

The dataset is constructed in a sound way.
The evaluation setup is fair and the results are reproducible.
The description of the dataset is correct.

**Documentation:**

The dataset and code base are hosted on GitHub and well documented. The maintenance plan is included in the Supplementary Material.

**Ethics:**

No. The author already discussed the problems of malicious code in appendix C.

**Relation To Prior Work:**

The paper clearly discussed how this work differs from previous contributions.

**Summary And Contributions:**

The paper introduces a new dataset of programming puzzles to benchmark the performance of AI models on program synthesis.

The paper makes the following contributions:
1. A novel problem representation called "Programming Puzzle".
2. A comprehensive dataset of programming puzzles that spans problems of a wide range of domains and difficulties.
3. A comprehensive benchmark of baseline systems including enumerative methods and GPT-3 solvers.
4. A user study to introduce a human baseline.

---

> ### Author Response · Authors · 2021-07-13
> **That you for your comments**
>
> Thank you for the helpful comments. Our response to individual comments is provided below. We will upload an updated version of the manuscript that addressed these points soon.
>
> * **For some domains (e.g., Algebra, ICPC, IMO) none of the baselines was able to solve any of the problems**: Indeed, our domains vary in difficulty and some are overall harder than others. For example, most of the problems in “trivial inverse” are relatively easy. As we discuss at the end of the second paragraph of section 2, solving some puzzles in P3 (e.g. open problems) is significantly more impressive than solving other, trivial problems. That said, our goal is to continue expanding the P3 dataset and to obtain a range of difficulties within each domain. Following this comment, we will first consider adding easier problems (that might be solved by current baselines).
>
> * **No human baseline for the whole dataset**: Our main goals in the human study were to (1) compare the difficulties for humans and AI solvers and (2) find whether puzzles are a reasonable (and enjoyable) measure of programming proficiency for humans as well. We also had to follow some constraints such as the total amount of time we ask participants to invest in the study. We found our designed study to meet our constraints while answering our research questions: (1) We found a positive correlation between the perceived puzzle difficulty for humans and AI systems and (2) participants with a range of experience levels provided positive feedback on the format and enjoyed solving the puzzles, and experienced coders performed better. As we plan to continue enlarging the dataset, we will also attempt to organize a contest to obtain additional statistics on other problems.
>
> * **Other related work**: Thank you for mentioning **CodeXGLUE**. We will include this collection of benchmarks in the related work section and their use of the CodeBLEU score that for better code generation evaluation. Our P3 dataset is different from all the tasks in that benchmark since we focus on problem-solving proficiency without the need for natural language descriptions. Also, our problems don’t require any answer-key. Regarding **Copilot**, the tool was released only after our submission and is still not publicly available. The [preprint](https://arxiv.org/abs/2107.03374) describing the underlying model was only released on July 7. We will add a citation to that work and we already reached out to the authors as we agree that this evaluation is interesting.

---

> ### Author Response · Authors · 2021-07-14
> **Updated paper and code for solvers**
>
> Thank you again for your helpful comments.
> We have updated the manuscript and added code for reproducing our main experiments to our repository: https://github.com/microsoft/PythonProgrammingPuzzles/tree/main/solvers
>
> Please let us know if you have any other comments.

---

### Author Response · Authors · 2021-07-13
**Revised manuscript following reviewers' comments**

We thank the reviewers for their valuable feedback. Following their comments and suggestions, we have updated the manuscript with the following modifications:

1. Figure 3(a) now reports the average performance across three runs.
2. Table 2 includes a score that is the macro-average success rate across different domains (giving equal weight for each domain).
3. Extended related work with the ARC challenge, CodeXGLUE, and Copilot.
4. Clarification regarding the use of RoBERTA (and link to the pretrained model).
5. Removed sentences about IQ test.
6. Footnote 5 - a disclaimer that the  Collatz conjecture is believed to be true.

---

### Author Response · Authors · 2021-07-14
**Added solvers to our repository**

We have added code for reproducing our main experiments to our repository: https://github.com/microsoft/PythonProgrammingPuzzles/tree/main/solvers

---

### Decision · Program_Chairs · 2021-07-26

**Decision:**

Accept

**Comment:**

In this paper the authors provide a novel dataset/task of "programming puzzles" where an AI is tasked with understanding a program and giving it an input that will make the program output "True".  All reviewers were very positive, in particular noting the novelty and generality of the dataset and task, as well as the thoroughness of the paper in having SOTA baselines. I believe this will be valuable for the academic community.